# Sarcoidosis: A Clinical Overview from Symptoms to Diagnosis

**DOI:** 10.3390/cells10040766

**Published:** 2021-03-31

**Authors:** Pascal Sève, Yves Pacheco, François Durupt, Yvan Jamilloux, Mathieu Gerfaud-Valentin, Sylvie Isaac, Loïc Boussel, Alain Calender, Géraldine Androdias, Dominique Valeyre, Thomas El Jammal

**Affiliations:** 1Department of Internal Medicine, Lyon University Hospital, 69007 Lyon, France; yvan.jamilloux@chu-lyon.fr (Y.J.); mathieu.gerfaud-valentin@chu-lyon.fr (M.G.-V.); thomas_3901@hotmail.fr (T.E.J.); 2Université Claude Bernard Lyon 1, Research on Healthcare Performance (RESHAPE), INSERM U1290, 69007 Lyon, France; 3Faculty of Medicine, University Claude Bernard Lyon 1, F-69007 Lyon, France; yves.pacheco@univ-lyon1.fr; 4Department of Dermatology, Lyon University Hospital, 69004 Lyon, France; francois.durupt@chu-lyon.fr; 5Department of Pathology, Lyon University Hospital, 69310 Pierre Bénite, France; sylvie.isaac@chu-lyon.fr; 6Department of Radiology, Lyon University Hospital, 69004 Lyon, France; 7Department of Genetics, Lyon University Hospital, 69500 Bron, France; alain.calender@chu-lyon.fr; 8Department of Neurology, Service Sclérose en Plaques, Pathologies de la Myéline et Neuro-Inflammation, Hôpital Neurologique Pierre Wertheimer, Lyon University Hospital, F-69677 Bron, France; geraldine.androdias-condemine@chu-lyon.fr; 9Department of Pneumology, Assistance Publique-Hôpitaux de Paris, Hôpital Avicenne et Université Paris 13, Sorbonne Paris Cité, 93008 Bobigny, France; dominique.valeyre@aphp.fr

**Keywords:** sarcoidosis, granulomatosis, cardiac sarcoidosis, neurosarcoidosis, diagnostics, differentials

## Abstract

Sarcoidosis is a multi-system disease of unknown etiology characterized by the formation of granulomas in various organs. It affects people of all ethnic backgrounds and occurs at any time of life but is more frequent in African Americans and Scandinavians and in adults between 30 and 50 years of age. Sarcoidosis can affect any organ with a frequency varying according to ethnicity, sex and age. Intrathoracic involvement occurs in 90% of patients with symmetrical bilateral hilar adenopathy and/or diffuse lung micronodules, mainly along the lymphatic structures which are the most affected system. Among extrapulmonary manifestations, skin lesions, uveitis, liver or splenic involvement, peripheral and abdominal lymphadenopathy and peripheral arthritis are the most frequent with a prevalence of 25–50%. Finally, cardiac and neurological manifestations which can be the initial manifestation of sarcoidosis, as can be bilateral parotitis, nasosinusal or laryngeal signs, hypercalcemia and renal dysfunction, affect less than 10% of patients. The diagnosis is not standardized but is based on three major criteria: a compatible clinical and/or radiological presentation, the histological evidence of non-necrotizing granulomatous inflammation in one or more tissues and the exclusion of alternative causes of granulomatous disease. Certain clinical features are considered to be highly specific of the disease (e.g., Löfgren’s syndrome, lupus pernio, Heerfordt’s syndrome) and do not require histological confirmation. New diagnostic guidelines were recently published. Specific clinical criteria have been developed for the diagnosis of cardiac, neurological and ocular sarcoidosis. This article focuses on the clinical presentation and the common differentials that need to be considered when appropriate.

## 1. Introduction

Sarcoidosis was first described by Besnier et al. in 1889 [1]. It is a multi-system disease of unknown etiology characterized by the infiltration of various organs by non-necrotizing granulomas. Even if sarcoidosis remains a disease of unknown cause, the mechanisms underlying granuloma formation are better and better understood, including genetic susceptibility and environmental factors [2].

Sarcoidosis can occur regardless of ethnicity or age. However, African Americans and Scandinavians have a higher incidence of the disease than the rest of the Caucasian population. Sarcoidosis generally starts in adults under 50 years of age. About 70% of cases occur between 25 and 40 years of age at presentation, with a second peak of incidence in women over 50 years old [3]. Its incidence is estimated to be between 2.3 and 11 per 100,000 individuals/year [4]. The estimated prevalence varies from 2.17 to 160 per 100,000 individuals. This high variability could be explained by the various diagnostic tools used in older series to define sarcoidosis and by the ethnicity of each cohort. In a well-conducted five-year study from a health maintenance organization in the United States (US), the age-adjusted annual incidence was 10.9 per 100,000 among Caucasian Americans and 35.5 per 100,000 for African Americans. The lifetime risk of sarcoidosis was estimated at 0.85% for Caucasian Americans and 2.4% for African Americans [5].

Sarcoidosis can follow two different courses: a time-limited one (in which two-thirds of the patients evolve through a self-remitting disease within 12 to 36 months) and a chronic course (in which 10 to 30% of patients require prolonged treatment) [3,4,6]. Not all patients with sarcoidosis will require systemic treatment, which is often reserved for life-threatening organ involvement (advanced pulmonary fibrosis, pulmonary hypertension, central nervous system (CNS) sarcoidosis, cardiac sarcoidosis, portal hypertension, etc.) or functional threat (severe or defacing skin disease, laryngeal involvement and/or posterior uveitis) [7]. Ethnicity (particularly African American and Afro Caribbean origins), age over 40 years at presentation, lupus pernio, chronic uveitis, sinonasal and osseous localizations, CNS involvement, cardiac involvement, severe hypercalcemia, nephrocalcinosis and radiographic stages III and IV have been associated with a poor prognosis [8]. Patients with sarcoidosis have a shorter life expectancy than the general population [9]. The mortality ratio in the sarcoidosis patient population can, for example, exceed 25 per million in African American women [10]. In Western countries, most sarcoidosis deaths are due to advanced pulmonary fibrosis leading to respiratory failure, pulmonary hypertension or both [11] and less commonly, cardiac and CNS sarcoidosis or portal hypertension [3]. In Japan, the main cause of mortality in sarcoidosis patients is cardiac involvement, which is responsible for 77% of deaths in people with sarcoidosis [12]. In addition to treatment-related mortality, less common causes of death include lymphoma and hemoptysis due to mycetoma [10,13].

## 2. Clinical Manifestations

Sarcoidosis is remarkable in that it can affect any organ. Common presentations are discussed below, with an estimate of their frequency; these estimates are based on data from US, Japanese and European studies [12,14,15]. In older studies, up to half of the patients could have asymptomatic disease, identified on chest X-ray (CXR) performed for other reasons. Fortuitous discovery on CXR is now very rare, reported in 8.4% of patients [16,17]. General symptoms like fatigue are seen in nearly 70% of patients, sometimes at the forefront of clinical features. More than 20% of patients have peripheral lymphadenopathy involving the cervical, axillary, inguinal and epitrochlear glands. The affected lymph nodes are moderately swollen and are usually painless. Differential diagnosis should include lymphoproliferative disorders such as Hodgkin’s lymphoma and infectious diseases including leishmaniasis, toxoplasmosis and tuberculosis [18].

### 2.1. General Symptoms

General symptoms are frequent in sarcoidosis [19]. For example, the prevalence of fatigue in sarcoidosis patients can reach 50 to 70% according to series [20]. Fatigue is not specific for sarcoidosis and it can be associated with different causes such as hypothyroidism, anxiety, depression, sleep apnea or active and severe inflammatory reaction [20]. A positive association was found between small fiber neuropathy and fatigue and also between dyspnea and fatigue [19]. The accurate detection of fatigue can be made with the Fatigue Assessment Scale [21]. The detection of fatigue is of utmost importance in sarcoidosis because on the one hand, fatigue is negatively related to quality of life in studies [22,23,24,25] and, on the other hand, the implementation of an adapted treatment can improve the patient’s symptoms [20]. Concentration disturbances are also a frequently reported symptom in patients with sarcoidosis and may be due to various associated comorbidities or consequences of sarcoidosis (e.g., sleep apnea and obstructive pulmonary disease). It is assumed that systemic treatment has a positive effect on sarcoidosis-associated cognitive complaint and cognitive disorders [19,26].

Other non-specific constitutional symptoms in sarcoidosis can include fever and weight loss. In most cases, fever remains low grade but can sometimes reach 39 to 40 °C [27]. In sarcoidosis, fever can be encountered during LS. Nevertheless, fever in a patient diagnosed with granulomatosis should rise the question of differential diagnoses and especially infectious differentials (e.g., tuberculosis). Sarcoidosis can also be a cause of fever of unknown origin and should be kept in mind when facing this clinical presentation [28]. Other general symptoms may be part of the initial clinical picture of sarcoidosis, such as weight loss and night sweats [27]. Of note, patients with hepatic manifestations of sarcoidosis can present with fever, night sweats and weight loss along with anorexia [29].

### 2.2. Lungs

Respiratory symptoms are found at presentation in 30–53% of patients; cough in 27–53%, dyspnea in 18–51% and chest pain in 9–23% [16,30,31]. Chronic dyspnea is most frequently seen in patients with a delayed diagnosis, such as in the 10% of patients diagnosed with sarcoidosis-related lung fibrosis [9]. Bilateral perihilar lymphadenopathy, which is most frequently mediastinal lymphadenopathy [32] and perilymphatic pulmonary nodules predominantly seen in the upper lobe are the most typical imaging findings. Scadding’s classification defines five stages of sarcoidosis on a CXR (Figure 1).

This system was developed prior to computed tomography (CT) and is widely used for its prognostic value. Mediastinal lymphadenopathy, especially right paratracheal and aorto-pulmonary locations, are commonly observed on chest CT. Calcifications [33] of lymph nodes may occur in sarcoidosis; they are usually chalky, focal and tend to be bilateral when present [33]. Chest CT is much more sensitive than CXR for the detection of lung nodules and subtle fibrosis. Pulmonary nodules tend to be tiny, usually termed “micronodules” ranging from 2 to 5 mm, typically located along the bronchovascular bundles, interlobular septa, interlobar fissures and subpleural regions, which constitute the “perilymphatic distribution (Figure 2A). Pulmonary fibrotic changes may be a dominant feature with typical features of architectural distortion, volume loss and bronchiectasis, which tend to predominate in the middle and upper lung zones (Figure 2B). Recently, the “dark lymph node” or the “cluster of black pearls” sign (defined by tiny round nodules each measuring 1–2 mm which are seen uniformly distributed throughout all or part of the lymph node) has been described as relatively specific of sarcoidosis with negative and positive predictive values of 96 and 91%, respectively [34]. The “galaxy” sign is also highly suggestive for sarcoidosis; it consists of a large nodule, usually with irregular boundaries, surrounded by a border of tiny satellite nodules. Alveolar, pseudo-alveolar consolidations, or diffuse ground glass are rarely the cause of sarcoidosis-associated radiological abnormalities [35].

Pulmonary function tests (PFTs) results generally correlate with the overall disease process assessed on lung histological examination between patients with mild and severe disease. However, PFTs do not differentiate the influence of alveolitis, granulomas or fibrosis [36]. PFT also reflect the impact of sarcoidosis at the pulmonary level but are not always correlated with radiographic staging (i.e., overlaps are not infrequent). The restriction of the lung volumes, particularly forced vital capacity (FVC), is the most common finding at spirometry and tends to be more frequent and marked from radiographic stage I to stage IV but with significant overlap at an individual level. Forced expiratory volume in one second (FEV1)/FVC ratio can be decreased [37] either in the case of significant bronchial distortion and stenosis due to pulmonary fibrosis [38] or to diffuse bronchial granulomatosis [39], proximal endobronchial stenosis [40,41], bronchial compression due to lymphadenopathy, granulomatous bronchiolitis or bronchial hyperreactivity [42]. The mechanisms involved are often multiple in the same patient and can be clarified by the combined use of CT, PFTs and bronchoscopy. Diffusing capacity of the lung for carbon monoxide (DLCO) is predictive of pulmonary vascular involvement when its value is more importantly decreased than that of FVC. Low DLCO value can also be the result of diffuse parenchymal involvement or alveolitis. PFTs have a prognostic value. The Composite Physiologic Index (CPI) is a composite score calculated with disease extent on CT, DLCO, FVC and FEV1 values and is more closely correlated to disease extent in idiopathic pulmonary fibrosis than isolated parameters from PFTs such as FVC and DLCO [43]. CPI is also more correlated to disease extent than FVC because of DLCO implementation in the calculated index. The CPI when >40 is predictive of mortality in sarcoidosis patients [44]. Six-minute walk test (6MWT) distance is often reduced and correlated with FVC decrease and results of St George’s respiratory questionnaire [22]. Cardiopulmonary exercise testing helps to understand the underlying mechanism of dyspnea of uncertain origin, particularly when spirometry and DLCO are within normal ranges [45,46,47,48,49].

As we discussed below, bronchoscopy with endobronchial and transbronchial lung biopsy as well as endobronchial ultrasound (EBUS) are cornerstones investigations for the histological diagnosis of sarcoidosis. Bronchoalveolar lavage (BAL) is a safe and minimally invasive procedure for the identification of the CD4+ alveolitis in sarcoidosis. The characteristic finding is a lymphocytic alveolitis in 80% of cases and T lymphocyte CD4/CD8 ratio over 3.5 in 50% of cases [50]. However, BAL lymphocytosis is not specific for sarcoidosis and the importance of the CD4/CD8 ratio is controversial unless it is greater than 3.5, showing a specificity of 93 to 96%. It is admitted that in inactive disease, the ratio is usually in the normal range. In most cases, BAL is not diagnostically decisive [36]. Eventually, the neutrophil count in BAL may increase in advanced sarcoidosis denoting the presence of pulmonary fibrosis and thus indicating an unfavorable prognosis [50]. High percentages of CD4 V2.3+ T cells (>10.5%) are associated with a better prognosis [51]. CD103+CD4+ T cells count as well as CD103+CD4+/CD4 in BALF was also found to be relevant for sarcoidosis diagnosis. A cutoff point of 0.45 (where abnormal ratio is considered under this value) was found to be associated with acceptable diagnostic performances (sensitivity: 81%; specificity: 78%) even in patients with normal CD4/CD8 ratio in BAL fluid [52]. Heron et al. found that combining CD4/CD8 and CD103+CD4+/CD4 ratio can be an specific diagnostic tool (specificity: 92%, positive predictive value: 93%) when considering respective cutoffs of 3 and 0.2, to differentiate patients with sarcoidosis from other interstitial lung diseases, even when CD4 alveolitis is missing [53].

The prevalence of pulmonary arterial hypertension (PAH) in sarcoidosis varies between studies depending on the characteristics of the study population and the methods used for the detection of PAH and its definition [54]. In studies of symptomatic patients or those waiting for lung transplantation, the prevalence of pre-capillary PAH, defined by mean pulmonary artery pressure (mPAP) >25 mmHg with pulmonary arterial wedge pressure (PAWP) <15 mmHg, is 5–74% [54]. In sarcoidosis, complex pathophysiological interactions may occur between the pulmonary vascular system and the parenchymal, mediastinal and cardiovascular compartments. The diagnosis can only be confirmed with right heart catheterization [7]. Elevated pulmonary pressure can be attributed to the granulomatous involvement of the pulmonary vessels or be the consequence of parenchymal destruction or compressive mediastinal infiltration. Thus, in sarcoidosis, PAH is a complication with functional and prognostic consequences [54].

### 2.3. Löfgren’s Syndrome

Löfgren’s syndrome (LS), first described in 1952 by Swedish Professor of Medicine Sven Löfgren, is a clinically distinct phenotype of sarcoidosis. Patients typically experience an acute onset of the disease, usually with fever (53.4%) and characteristic symptoms consisting in bilateral hilar lymphadenopathy, erythema nodosum (Figure 3) (57–100%) and/or bilateral ankle arthritis or periarticular inflammation (81.2%) [55]. LS occurs mostly in European Caucasians, especially in Sweden and in the Netherlands where LS patients constitute roughly a third of all sarcoidosis cases. It is less common in the United Kingdom and in the US, where only 0.9 and 0.7% of sarcoidosis patients present with LS, respectively, and is extremely rare in Asia. It usually occurs between the age of 25 to 40, with a second peak around the age of 40 to 60 and is more frequent in women (70%) [56]. The different manifestations of LS differ according to sex. Erythema nodosum is found predominantly in women while arthropathy/arthritis is more common in men. In Löfgren’s original cohort as well as in more recent studies, most patients (79–82.5%) present with radiographic stage I, 14.6–23% have stage II disease while no patient present with stage III/IV [55,56]. On average 2% of the patients have no radiographic abnormalities. Magnetic resonance imaging (MRI) of the joints or ultrasonography typically shows periarticular inflammation with subcutaneous and soft tissue edema accompanied by small amounts of joint and tenosynovial fluid without evidence of synovial thickening or synovitis [57,58]. LS patients herald a benign and self-remitting disease, which is especially true for individuals carrying the HLA-DRB1*03 allele. Chronic trend >2 years occurs in 8 to 22.6% of LS patients and is associated with older age, stage II at diagnosis and the need for treatment [55]. Apart from erythema nodosum, fever, articular involvement and extrapulmonary manifestations have been reported in only 12% of LS patients [16]. Among them, uveitis, parotitis, facial palsy, skin (except lupus pernio), liver or spleen involvement have been described as more frequently associated with LS.

### 2.4. Musculoskeletal Manifestations

Joint involvement, also known as sarcoid arthropathy, is observed in 6–35% of patients and asymptomatic bone involvement occurs in 3–13% of patients [59]. Other manifestations include sarcoid myopathy (<3%) and hypercalcemia (around 6%) [60].

Chronic sarcoid arthritis is less frequent (7%) than LS and is also known to commonly affects the ankles [61]. It is characterized by persistent (>3 months), symmetrical, oligo or polyarthritis of the medium and large joints in 20% of patients, with 40% arthralgia [62]. It is important to distinguish true synovitis from tenosynovitis, as the latter is more frequent [60]. Chronic forms have a less symmetrical joint and skin involvement distribution than LS, whereas ocular involvement is more frequent [61]. They are usually associated with parenchymal lung or extra articular sarcoidosis [60,63]. Polyarthritis only affecting the small joints of the hands is very rare and should be first considered as another systemic arthritis, such as rheumatoid arthritis, which may even coexist with sarcoidosis. Erosive changes and Jaccoud’s-type arthropathy have been described in case-reports [64]. Erb et al. reported a 6% prevalence of spondyloarthritis in patients with sarcoidosis suggesting a possible association between the two conditions [65].

Bone sarcoidosis is observed in 0.5% to 30% of patients depending on the sensitivity of imaging procedures and is frequently associated with lupus pernio, uveitis and a chronic multisystemic course of the disease [60]. On the other hand, Sparks et al. reported only 20 cases (1.5%) of osseous involvement detected in 2013 patients with sarcoidosis identified between 1994 and 2013 at Brigham and Women’s Hospital in Boston, Massachusetts [66]. Bone lesions are asymptomatic in half cases. They are commonly located in the phalanges of the hands and feet and are usually bilateral. Dactylitis typically involves the bones and soft tissue with slight swelling, tenderness and limitations of the movement (Figure 4A). The skull, as well as long bones, ribs, pelvis and axial skeleton may also be affected. Radiographic investigations may revelate sclerotic (typically in the spine) and/or osteolytic lesions, cystic and punched out lesions and cortical abnormalities (Figure 4B). Vertebral sarcoidosis can affect lower thoracic and upper lumbar vertebrae and may mimic metastatic lesions. The use of more sensitive imaging procedures, such as MRI or nuclear imaging, reveals a higher rate of bone involvement than does radiography [67,68]. Using fluorine-18 fluorodeoxyglucose positron emission tomography/computed tomography (^18^F-FDG PET CT), Demaria et al. have recently shown that bone sarcoidosis occurs in 14% of patients with sarcoidosis and affects multiple bones and mostly the axial skeleton (spine then pelvis, then sternum). Only patients with hand involvement had bony symptoms (pain and/or swelling) [69].

Asymptomatic granulomatous muscle involvement in sarcoidosis has already been reported, with a prevalence of 50–80% [70], whereas symptomatic muscle involvement is less common (1.4–2.3%) [71,72]. Symptomatic involvement may include a palpable nodular type, which is infrequent; an acute myositis type, which is rare and seen more commonly in early sarcoidosis; and a chronic sarcoid myopathy type, which is more common, with a slower onset and occurs later in life [73]. Diagnostic difficulty may arise when the myopathy occurs in patients ongoing corticosteroid therapy, since corticosteroid-associated myopathy can present in a very similar manner to what is seen in sarcoid myopathy. As with bone sarcoidosis, ^18^F-FDG PET CT has emerged as a sensitive imaging technique for muscle involvement in sarcoidosis [74]. Most patients with acute sarcoid myopathy improve under systemic treatment, whereas chronic sarcoid myopathy is frequently associated with severe disability and rarely improves after corticosteroid treatment, immunosuppressive or anti-tumor necrosis factor (TNF)-α [75,76]. In a recent study, including 23 patients with granulomatous myositis, Dieudonné et al. reported that almost half of the patients met the criteria for sporadic inclusion body myositis which was the only factor associated with unresponsiveness to treatment in patients with granulomatous myositis [77].

Depending on the studies and the study population, hypercalcemia affects 7–18% of patients with sarcoidosis [12,78,79,80]. In a case–control etiologic study of sarcoidosis named ACCESS, a multicenter prospective study with 736 enrolled patients, the incidence of sarcoidosis-associated hypercalcemia at onset was 3.7%. The largest up-to-date study, a single-center retrospective study by Baughman et al. (n = 1606), reports that hypercalcemia appears in about 6% of sarcoidosis patients [81]. The majority of hypercalcemia in sarcoidosis are explained by the ectopic production of 1,25(OH)_2_D_3_ (calcitriol) by activated macrophages within granulomas. A more frequent sign of dysregulated calcium homeostasis in sarcoidosis is hypercalciuria which affects 20–40% of patients. As a consequence, nephrolithiasis is more frequent in sarcoidosis than in the general population as it occurs in 10–14% of patients during the course of the disease [82]. It is important to remember that sarcoidosis does not exclude other causes of hypercalcemia, starting with the most important: hyperparathyroidism and neoplasia, particularly lymphoma which can be associated with increased calcitriol production in granulomas [83].

### 2.5. Skin

According to studies, skin is the second or third most commonly affected organ in sarcoidosis, present in up to one-third of the patients [84]. Cutaneous sarcoidosis lesions are frequently the inaugural signs of the disease and sarcoidosis can remain an isolated dermatological condition in more than 30% of cases [85]. In patients with cutaneous and systemic sarcoidosis, skin findings rise before or at the time of diagnosis in 80% of patients [85]. Recognizing sarcoidosis cutaneous manifestations is of most importance as they are frequently the first sign of the disease and because skin is an easily accessible tissue for biopsy. There are “specific” and “non-specific” findings based on the presence or absence of characteristic sarcoidosic granulomas on histological examination.

There is a wide variety of sarcoidosis-specific skin lesions, some more common than others (Table 1). Skin manifestations are typically multiple erythematous macules, papules, plaques or subcutaneous nodules. They usually do not cause symptoms but they can be of aesthetic importance when localized to the face, as in the classical lupus pernio, which consists of indolent violaceous indurated plaques usually affecting the nose, cheeks and ear lobes. Specific cutaneous sarcoidosis may also occur in scar tissue, at traumatized sites and around embedded foreign bodies, such as in tattoos. Cutaneous sarcoidosis is often considered as a great mimicker. Psoriasiform, lichenoid, verrucous and angiolupoid are less frequently seen variants of papular or plaque sarcoidosis that can be confused with psoriasis, lichen planus, warts or lupus erythematosus, respectively. Some patterns of cutaneous involvement may be associated with specific extracutaneous manifestations of sarcoidosis, while other patterns may predict systemic disease severity and response to treatment. The clinical appearance may vary based on the morphological type of the lesion, its chronicity and the color of the surrounding skin (Figure 5). Erythema nodosum is the most common non-specific lesion, developing in up to 25% of cases [86]. Histological examination is not warranted since histological structure of erythema nodosum is not specific to sarcoidosis and never granulomatous.

Erythema nodosum is mostly seen in LS and is associated with transient disease that does not require treatment. Other non-specific cutaneous findings of sarcoidosis include calcinosis cutis, clubbing and prurigo [84].

### 2.6. Eyes

The reported prevalence of ocular involvement ranges from 10% to 50%, with a higher prevalence observed in African Americans than Caucasians and among women than men (female to male ratio of about 2:1). Ocular sarcoidosis may develop in the absence of any apparent systemic involvement or may be the main site of the disease without significant clinical disease elsewhere.

All ocular structures may be involved (Table 2), but uveitis is the most frequent form of ocular manifestation and may affect up to 20–30% of patients with sarcoidosis [87]. Apart from uveitis, lacrimal-gland enlargement (which can induce sicca syndrome) and conjunctiva involvement are the most frequent features while optic neuritis is challenging for the clinician because of its severity that often requires systemic treatment.

Sarcoid uveitis is generally bilateral (75–90%) with the same findings and clinical course in both eyes [88,89]. Anterior uveitis (defined as iritis and/or iridocyclitis) is the most common anatomical form of intraocular inflammation (41–75% of sarcoid uveitis) [88,90,91], followed by posterior, intermediate uveitis and panuveitis. Nevertheless, recent reports from tertiary centers identified panuveitis as the most commonly encountered subtype [89,92]. Anterior uveitis is usually chronic (defined as a relapse within 3 months after treatment discontinuation) [90], bilateral and granulomatous (Figure 6). Uveoparotid fever, also known Heerfordt(–Waldenström)’s syndrome is a highly specific subtype of sarcoidosis [93,94]. It is characterized by a combination of facial palsy, parotid gland enlargement, uveitis and is associated with low-grade fever [95]. Cases that manifest with all of the three symptoms are called “complete Heerfordt’s syndrome” (0.3% of sarcoidosis patients). If only two of the three characteristic symptoms are present, it is called” incomplete Heerfordt’s syndrome” (1.3%). Heerfordt’s syndrome can be rarely associated with cranial nerve involvement, particularly affecting the trigeminal nerve [96] along with visceral involvement [97].

Sarcoid uveitis is associated with a favorable visual outcome, since most patients experience mild or no visual impairment [98,99]. However, 2.4 to 10% of patients with sarcoid uveitis develop severe visual impairment (defined as best-corrected visual acuity <20/200 in at least one eye) [90,92,98,100] with cystoid macular edema being the main cause of visual loss [98,101,102]. Ocular inflammation related to sarcoidosis can have a smoldering course and patients can remain asymptomatic for a long time [103]. Therefore, ophthalmologic screening is recommended for all patients with newly diagnosed sarcoidosis, even in the absence of symptomatic ocular sarcoidosis.

### 2.7. Liver, Spleen and Gastrointestinal Involvement

Autopsy series report hepatic involvement in up to 80% of cases [108]. However, asymptomatic elevation in liver-function tests in the context of known sarcoidosis is the most common presentation in approximately one-third of patients and occurs more frequently in African Americans than in Caucasians [109]. Of note, alkaline phosphatase seems to be more consistently elevated than aminotransferases in patients with hepatic sarcoidosis. Clinical manifestations include hepatomegaly, fatigue, right upper quadrant abdominal pain with pruritus (5–15%), fever, jaundice and weight loss (less than 5%) [110]. Portal hypertension has been reported in 3 to 20% of cases [109,111] of sarcoid hepatitis and can be secondary to: (i) the obstruction of portal venous system due to large granulomas in the portal areas, causing a presinusoidal block; (ii) ischemia secondary to primary granulomatous secondary to granulomatous phlebitis of the portal and hepatic veins causing cirrhosis and focal fibrosis [112], or (iii) by arteriovenous shunts that increase portal blood flow [112]. Rare complications include Budd–Chiari syndrome, cholestatic liver disease mimicking primary biliary cirrhosis, primary sclerosing cholangitis, or obstructive jaundice [113,114]. CT scan and ultrasonography of the liver identify abnormalities in about half of the patients, with hypodense nodules being the most commonly observed (5–35%), followed by hepatomegaly (8–18%) [115,116,117]. MRI and ^18^F-FDG PET CT provide the best resolution and are the most sensitive modality for detecting liver nodules. The definitive diagnosis of hepatic sarcoidosis requires detection of non-caseating granuloma in the liver and exclusion of other diseases, such as alcoholic hepatitis, nonalcoholic liver fatty disease, infections and drug-induced liver injury. These granulomas are commonly focal and are found in the portal and periportal areas (although they can be found in all lobules) [118,119]. Common causes of granulomatous hepatitis include infections (e.g., *Mycobacteria* spp., *Yersinia* spp., *Coxiella burnetii*, *Toxoplasma gondii*, *Bartonella henselae*, *Brucella* spp., hepatitis C virus), immunological disorders (e.g., primary biliary cirrhosis, Crohn’s disease), environmental microparticles exposure (e.g., beryllium), neoplasms and drug reactions [120]. In the Western world, the two main causes of granulomatous hepatitis are primary biliary cirrhosis and sarcoidosis [121,122,123]. Inflammation of the liver can persist and cirrhosis can develop in a minority of patients (3–6%) [124].

Splenic involvement is most often detected by imaging rather than by symptoms or laboratory abnormalities and has been reported with a similar frequency as hepatic involvement. It may present with constitutional symptoms and marked splenomegaly in up to 6% of cases [116,125]. CT show spleen enlargement or splenic nodules which can be observed in up to 15% of abdominal CTs, as single or multiple hypodense lesions larger than 1 cm, with an irregular shape and a confluence tendency. Punctate, hyperdense calcifications have been found in up to 16% of the patients and also more clearly calcified lesions have been reported [126,127]. A recent study showed that diffuse splenic involvement was associated with an extensive extrapulmonary course and diffuse endobronchial disease and appears to be a risk factor for persistent chronic sarcoidosis [128].

An enlargement of the abdominal lymph nodes is present in about 30% of patients and is mainly located in the hepatic hilum and at the para-aortic and celiac sites, around iliac vessels or in the mesentery. The lesions usually appear hypodense with size ranges from 1 to 2 cm. Lymph nodes greater than 2 cm are observed in up to 10% of patients, which raise the problem of differential diagnosis with malignant lesions such as lymphoma [126,129,130].

Gastrointestinal tract involvement is extremely rare, described in 0.1 to 1.6% of cases [131]. Gastric or intestinal sarcoidosis presents with abdominal pain, weight loss, nausea, vomiting, protein-loss enteropathy and signs of gastrointestinal obstruction or bleeding. These findings are also seen in Crohn’s disease [132,133]. On endoscopy, macroscopic lesions are observed in the esophagus (9%), stomach (78%), duodenum (9%), colon (25%) and rectum (19%). As compared with Crohn’s disease, digestive tract sarcoidosis is associated with Afro-Caribbean origin and the absence of ileal or colonic macroscopic lesions [131].

### 2.8. Cardiac Sarcoidosis

Cardiac involvement affects approximately 3 to 39% of patients with systemic sarcoidosis [134,135,136,137,138,139,140]. However, autopsy series of patients with systemic sarcoidosis reported cardiac granulomas in a higher proportion (up to 46.9% of cases) [141,142]. There is a predilection for the left ventricular wall, the interventricular septum and the conducting system but any part of the heart can be affected [143]. Patients with cardiac sarcoidosis (CS) are mostly asymptomatic, but may present with chest pain, palpitations, dyspnea, congestive heart failure, pericardial effusion or syncope/presyncope due to arrhythmias. The most common abnormality is atrioventricular block (45%) [144]. For this reason, any conduction defect on electrocardiogram (bundle branch block, prolonged PR interval) or other nonspecific changes should prompt further cardiological investigations. Pathological Q waves (pseudo-infarct pattern), ST segment and T wave changes and rarely epsilon waves can occur [138]. Current ATS guidelines recommend that electrocardiogram should not be performed every year in any patient with known sarcoidosis since clinically silent CS usually follows a benign course [27,145].

Other classical manifestations include arrhythmia (atrial or ventricular (35.6%) [146] and cardiomyopathy leading to heart failure (10–20% of patients). Less commonly, CS may manifest as pericardial, valvular or coronary heart disease. Transthoracic echocardiography (TTE) sensitivity in CS is around 25% [139,140]. Interventricular thinning (particularly basal interventricular thinning) is the most typical feature of CS and has been shown to be highly specific for diagnosis. Other features include increased myocardial wall thickness, ventricular aneurysms, left ventricular and/or right ventricular diastolic and systolic dysfunction and isolated wall movements abnormalities [147].

TTE is a good screening tool for diagnosis when facing respiratory or cardiac symptoms in sarcoidosis, making it possible to rule out several differential diagnoses, such as pulmonary hypertension, valvular disease, or ischemic heart disease. Of note, a normal TTE does not exclude CS.

Cardiac MRI (CMRI) is the cornerstone for the diagnostic work-up of CS given its sensitivity and specificity (both over 90%), by identifying areas of myocardial damage including edema and scarring, primarily through late gadolinium enhancement (LGE) [138,140,148]. LGE is typically multifocal and patchy and is usually seen in the epicardium and mid-myocardium. The prognostic value of LGE on CMRI is equally important and allows risk stratification in cardiac sarcoidosis, since LGE is associated with death and ventricular tachycardia as well as the size of the granulomatous infiltrate which is associated with a poor prognosis [140,149,150,151,152].

^18^F-FDG PET CT has a fair diagnostic accuracy for CS, with a meta-analysis reporting a pool sensitivity of 89% and specificity of 78% [153]. Considering such evidence, the latest guidelines have positioned CMRI above TTE and ^18^F-FDG-PET/CT for the confirmation of suspected CS in patients with extracardiac sarcoidosis [27]. Serial PET imaging is useful in monitoring disease activity and response to immunosuppressive therapy [154].

Endomyocardial biopsy has a low sensitivity (∼30%) given the patchy nature of cardiac sarcoidosis and is an invasive procedure [155].

### 2.9. Neurosarcoidosis

Neurological involvement in sarcoidosis is relatively uncommon, with a reported prevalence of 3 to 10% [156]. Isolated neurosarcoidosis is very rare (1–17%), with 84 to 94% of cases experiencing coexisting systemic manifestations of sarcoidosis, especially in the lungs and intrathoracic lymph nodes [157]. In approximately half of the cases of patients with neurosarcoidosis, neurological symptoms are the first manifestation that leads to the identification of the disease [158].

Every part of the nervous system can be affected and multiple parts can be affected at the same time, some more commonly than others (Table 3) [159,160]. The most frequently affected sites are the cranial nerves (55%), the meninges (12–40%), the brain parenchyma (20–45%) and the spinal cord (18–26.5%) [161]. The involvement of the pituitary gland (13.7%), peripheral nerves (10.3 to 17%) or stroke (2.6%) are less common [156,162]. All cranial nerves can be affected in neurosarcoidosis, with cranial nerves II, VII and VIII being the most commonly involved. On imaging studies, leptomeningeal abnormalities are three times more frequently detected than pachymeningeal disease [158].

Cerebrospinal fluid (CSF) angiotensin converting enzyme (ACE) had been proposed as a diagnostic tool with a relatively good specificity (67.3–95%) [163]. However, its sensitivity is low, ranging from 0 to 66.7% [156,164]. The CD4/CD8 ratio in CSF is higher in neurosarcoidosis than in multiple sclerosis and other neurological inflammatory disorders and may be helpful for the differential diagnosis [165]. Hypoglycorachia (up to 14%) and oligoclonal bands (up to 42%) are also biological features associated with neurosarcoidosis [156]. The differential diagnosis of intraparenchymal lesions is quite broad, including life-threatening conditions such as infections (e.g., tuberculosis), tumor (e.g., lymphoma) or other CNS inflammatory disorders (e.g., multiple sclerosis or neuromyelitis optica). Therefore, histopathological confirmation is generally necessary, although biopsy of those lesions is often technically challenging, with a relatively high rate of false-negative results (up to 40% in the Stern et al.’s study) [166].

Spinal cord sarcoidosis (SCS) occurs in less than 1% of all patients [167,168,169]. In comparison with other myelopathies, neurologic pain seems to be more frequent and may be considered as an alarm for early diagnosis of SCS [170]. Sarcoidosis associated myelitis is usually extensive (more than three vertebral levels) with a subacute/progressive onset compared with other myelopathies (vascular, inflammatory, or infectious myelopathies) (Figure 7). A distinct MRI phenotype, with enhancement in subpial and/or meningeal areas, also called “trident sign”, is seen in sarcoidosis-associated myelopathy and can help the diagnosis [171].

Hypothalamo-pituitary (HP) involvement is a rare manifestation of sarcoidosis, representing less than 1% of all intrasellar lesions [172]. Patients with HP sarcoidosis have significantly more frequent sinonasal and neurological involvement and more frequently require systemic treatment [173].

Peripheral neuropathy is rare in sarcoidosis. Nerve biopsy is often required to confirm the diagnosis. Small fiber neuropathy (SFN) is increasingly recognized as another manifestation of neurosarcoidosis and seems to be quite common with a reported prevalence of up to 10% of all patients with sarcoidosis in a retrospective study [174]. SFN symptoms can be disabling and severely impact quality of life [19]. Symptoms associated with SFN include pain and autonomic disorders such as dry eyes or mouth, orthostatic hypotension and urinary dysfunction [175]. Its detection is essential as sarcoidosis treatments may improve SFN-associated symptoms [176]. Classic electrophysiological tests usually fail to assess small fibers damages and indirect testing such as sudomotor testing or more specific investigations such as intraepidermal nerve fiber density in skin biopsy. Hoitsma et al. provided a diagnostic tool to detect small fiber neuropathy in sarcoidosis patients consisting of a brief and reproductible questionnaire [177].

### 2.10. Kidney Involvement

Granulomatous interstitial nephritis is the classic renal pattern of sarcoidosis, reported in up to 13% of patients in autopsy studies [183]. However, clinically evident interstitial nephritis is quite rare and is observed in 0.7 to 4.3% in clinical studies, suggesting a non-aggressive course of renal disease [12,15,184,185]. Except the patients diagnosed on autopsy, impairment of renal function with or without abnormal urinalysis results (microscopic hematuria (21.7%), aseptic leukocyturia (28.7%) and moderate proteinuria (66%)) is the most common presentation [183]. Most of them have experienced extrarenal sarcoidosis, with a higher frequency of hypercalcemia (32%) and fever (17%) at presentation [183,186]. A total of 19% to 21.3% of renal biopsies reveal only signs of interstitial nephritis without granuloma, which may reflect the inflammatory process or biopsy sampling error [183,186]. However, noncaseating granulomatous interstitial nephritis is not pathognomonic of renal sarcoidosis. Several diseases, such as tuberculosis, *Mycobacterium avium* infection, leprosy, fungal infections, foreign body reactions, inflammatory diseases (tubulointerstitial nephritis with uveitis syndrome, granulomatosis with polyangiitis and Crohn’s disease), crystal nephropathies and drug allergies can give rise to the same histopathologic picture [187,188,189].

Other renal diseases can be associated with sarcoidosis like nephrocalcinosis, nephrolithiasis secondary to hypercalcemia and hypercalciuria (see musculoskeletal manifestations section). Clinical manifestations are similar to those of nephrocalcinosis and nephrolithiasis from other causes [190].

Finally, glomerular diseases (membranous nephropathy, focal segmental sclerosis, mesangial proliferative glomerulonephritis, IgA nephropathy and extra capillary/crescentic glomerulonephritis) and isolated tubular dysfunction have been more rarely be reported [191].

### 2.11. Otolaryngological Involvement

Approximately 10–15% of the patients have symptomatic specific otolaryngological involvement of the larynx (0.5–1.4%), the major salivary glands, including Heerfordt’s and Mikulicz’s syndromes (5–10%) and the nose and paranasal sinuses (1–4%) [192,193]. While major salivary gland involvement most frequently follows a benign course, sinonasal and laryngeal sarcoidosis are usually severe, associated with other serious manifestations, with a particularly longstanding course and represent a therapeutic challenge. Patients with sinonasal disease have non-specific symptoms including: nasal obstruction (90%), rhinorrhea (70%), anosmia (70%), crusting rhinitis (55%), epistaxis (30%) and facial pain (20%) [193]. Based on the clinical presentation, Lawson et al. have classified sinonasal sarcoidosis into four subgroups: atrophic, hypertrophic, destructive and associated with nasal enlargement [194]. Laryngeal sarcoidosis usually involves the supraglottis (epiglottis, then arytenoids) and does not affect the vocal cords [195]. The most common symptoms are hoarseness (three-quarters of patients), inspiratory dyspnea (38–90%) and dysphagia (38–46.7%) [196,197]. It is often associated with other loco-regional localizations as lupus pernio and sinonasal involvement [196].

## 3. Clinical Phenotypes

In recent years, three studies have investigated the heterogeneity of sarcoidosis in order to identify clinical phenotypes with similar combinations of traits. In 2018, the GenPhenReSa project, involving 2163 Caucasian patients recruited from pulmonology departments, proposed five clinical clusters: (1) abdominal organ involvement, (2) ocular–cardiac–cutaneous–central nervous system involvement, (3) musculoskeletal–cutaneous involvement, (4) pulmonary and intrathoracic lymph node involvement and (5) extrapulmonary involvement [135].

Using cluster analysis in a cohort of 694 patients seen at their university hospital, Rubio-Rivas and Corbella identified six discrete phenotype subgroups: C1 (pure LS with bilateral hilar lymphadenopathy (BHL) and erythema nodosum [EN]), C2 (febrile LS), C3 (non-febrile LS with periarticular ankle inflammation), C4 (exclusive pulmonary sarcoidosis), C5 (pulmonary sarcoidosis and “abdominal” involvement) and C6 (organ involvement different from the lungs, including: skin lesions, peripheral lymph nodes and neurological and ocular involvement). Contrary to the first three subsets, most patients with C4-6 phenotypes were treated with immunosuppressive therapy and evolved to chronicity to a greater extent. More recently, the EpiSarc study analyzed 1237 patients from 11 French hospital centers [198]. The hierarchical cluster analysis identified five distinct phenotypes according to organ involvement, disease type and symptoms: (1) (n = 180) LS; (2) (n = 137) eye, neurological, digestive and kidney involvement; (3) (n = 630) pulmonary involvement with fibrosis and heart involvement; (4) (n = 41) lupus pernio and a high percentage of severe involvement; and (5) (n = 249) hepatosplenic, peripheral lymph node and bone involvement. Phenotype 1 was associated with European origin, female sex and with non-manual work; phenotype 2 with European origin; and phenotypes 3 and 5 with being non-European. The proportion of labor workers was significantly lower in phenotype 5 than in the other phenotypes. Altogether, these results suggest that these distinct phenotypes are sequelae of different etiological agents causing sarcoidosis that are either inhaled, ingested or passed through the skin, eyes and/or the blood-brain barrier [199]. In addition, different genetic risk profiles may predispose to these different phenotypes. Several studies are ongoing to associate clinical phenotypes with biological pathways and specific genotypes [135].

## 4. Diagnosis

The diagnosis of sarcoidosis is based on three major criteria: consistent and adequate clinical presentation; demonstration of the presence of non-caseating granulomas in one or more tissue samples; and the exclusion of other causes of granulomatous disorders [27]. In order to establish uniform standards for the probability of organ involvement in sarcoidosis, consensus criteria were originally established in 1999 [3], then updated in 2014 [200] by the World Association of Sarcoidosis and Other Granulomatous Disorders (WASOG), according to a Delphi study. Clinical manifestations were assessed as either:highly probable: likelihood of sarcoidosis causing this manifestation of at least 90% (e.g., uveitis, bilateral hilar adenopathy, perilymphatic nodules on chest CT);probable: likelihood of sarcoidosis of 50–90% (e.g., seventh cranial nerve paralysis, lachrymal gland swelling, upper lobe or diffuse infiltrates);Possible: likelihood of sarcoidosis of less than 50% (e.g., arthralgias, localized infiltrate on CXR) [200].

In 2018, using the WASOG organ involvement criteria, Bickett et al. proposed a Sarcoidosis Diagnostic Score which might accurately differentiate sarcoidosis from other granulomatous diseases [201]. An update of the international guidelines for the diagnosis and management of sarcoidosis published in 1999, was released in April, 2020 [27]. These guidelines have been developed according to the GRADE (Grading of Recommendations Assessment, Development and Evaluation) based on a systematic review of the literature and, where appropriate, meta-analysis, in order to summarize the best available evidence. The three recommendations about diagnosis were related to pathomorphological examination of the lymph nodes.

While many sarcoidosis cases constitute a diagnostic dilemma, certain clinical features are considered to be highly specific of the disease. These include LS, lupus pernio and Heerfordt’s syndrome (See above). In these settings, the American Thoracic Society (ATS) experts suggest that lymph nodes should not be collected (conditional recommendation, very low quality evidence).

A total of 16 studies were conducted in 556 asymptomatic patients with suspected radiographic stage 1 sarcoidosis and showed that sarcoidosis was confirmed in 85% of patients, while an alternative diagnosis was made in 1.9% (i.e., tuberculosis in 38% of cases and lymphoma in 25% of cases). Sampling was non-diagnostic in 11% of cases. Based on these data, experts make no recommendation for or against obtaining a lymph node sample in asymptomatic patients whose changes in the radiological image indicating bilateral hilar lymphadenopathy. The need for closer clinical examination is emphasized in patients whose biopsy was postponed. The third recommendation, for patients with suspected sarcoidosis along with hilar/mediastinal lymphadenopathy, is to perform Endobronchial Ultrasound (EBUS)-guided lymph node sampling, rather than mediastinoscopy (conditional recommendation, very low quality). Agrawal et al. recently performed a systematic review and meta-analysis which yielded, for sarcoidosis, a pooled overall diagnostic yield of 93% for EBUS-intranodal forceps biopsy vs. 58% for EBUS-transbronchial needle biopsy (*p* < 0.00001) [202]. Figure 8 provides an algorithm based on recent ATS guidelines for the diagnosis of sarcoidosis and that incorporates the value of minor salivary gland biopsy (MSGB) and ^18^F-FDG PET in sarcoidosis diagnosis [27,36,99]. Two studies in uveitis patients showed that granulomas were only found on MSGBs in patients with elevated ACE or with a compatible chest CT [203,204].

In case of poorly accessible organ localization without superficial nor thoracic manifestation (e.g., cardiac or neurosarcoidosis) ^18^F-FDG PET may show a suggestive sarcoid-like uptake pattern with hypermetabolic mediastinal and hilar lymph nodes, whether or not combined with lung parenchymal active disease which supports the likelihood of sarcoidosis and orientates a mediastinoscopy [205]. ^18^F-FDG PET may sometimes reveal smoldering superficial localizations, for example, cervical lymph nodes which are easily accessible to biopsy (Figure 9) [206]. The value of EBUS-guided lymph node sampling in patients with hypermetabolic lymph nodes on ^18^F-FDG PET remains to be studied.

### 4.1. Specific Diagnostic Criteria in Cardiac, Neuro and Ocular Sarcoidosis

#### 4.1.1. Cardiac Sarcoidosis

Due to the current limitations of endomyocardial biopsy as a reference standard, physicians mainly rely on advanced cardiac imaging, multidisciplinary evaluation and specific diagnostic criteria to diagnose cardiac sarcoidosis. The revised Japanese Ministry of Health and Welfare diagnostic criteria (Table 4) and the Heart Rhythm Society Expert Statement Criteria (Table 5) are currently the two main well-established diagnostic criteria guidelines for cardiac sarcoidosis [138,148]. Nevertheless, these two sets of diagnostic criteria were mainly based on expert consensus and have not yet been validated by prospective data or clinical trials [207].

#### 4.1.2. Neurosarcoidosis

An accurate diagnosis of neurosarcoidosis can also be challenging, as it requires histologic confirmation of the affected site and neural tissue is not easily accessible for pathologic examination. The Neurosarcoidosis Consortium Consensus recently proposed diagnostic criteria for central nervous system and peripheral nervous system neurosarcoidosis (Table 6) [209].

#### 4.1.3. Ocular Sarcoidosis

Biopsy is unacceptable for many patients with suspected sarcoidosis and uveitis. Therefore, a first International Workshop on Ocular Sarcoidosis (IWOS) criteria for the diagnosis of intraocular sarcoidosis have been published [210]. These criteria include a combination of suggestive ophthalmological findings and laboratory investigations when biopsy is not performed or negative. To overcome its low sensitivity, the revised IWOS criteria were recently established in an international consensus [211]. The survey and subsequent workshop reached consensus agreements on four criteria which are summarized in Table 7 [211]. The most substantial changes were the addition of four criteria: (1) lymphopenia; (2) CD4 alveolar lymphocytosis; (3) parenchymal lung changes consistent with sarcoidosis; and (4) abnormal label uptake on ^67^Ga scintigraphy or ^18^F-FDG PET CT.

### 4.2. Histopathology

Since the clinical manifestations of sarcoidosis are frequently non-specific, histological evidence of granulomas is often required to establish an accurate diagnosis. Sarcoidosic granulomas are composed of tightly clustered epithelioid histiocytes and occasionally multinucleated giant cells with few lymphocytes and often surrounded by fibrosis. An outer layer of loosely organized lymphocytes, mostly T cells, is often observed accompanied with few dendritic cells (Figure 10). In some cases, granulomas are surrounded by isolated collections of B-cells. Sarcoidosis granulomas are most of the time non-necrotizing. However, variants of sarcoidosis, particularly the nodular pulmonary sarcoidosis phenotype, can present with a mixture of necrotic (focal and usually minimal ischemic necrosis) and non-necrotic granulomas [212]. The differential diagnosis of granulomatous diseases is broad, as noted in the next section. Some histopathological features are not suggestive of sarcoidosis (e.g., few granulomas, loosely organized collections of mononuclear phagocytes/multinucleated giant cells, extensive necrosis, dirty necrosis (containing nuclear debris), palisading granulomas, lack of lymphatic distribution of granulomas, robust surrounding inflammatory infiltrate (including lymphocytes, neutrophils, eosinophils and plasma cells) andsecondary lymphoid follicles, although histopathologic features alone cannot distinguish sarcoidosis from other granulomatous diseases. Some granulomatous diseases may have similar histological features, such as berylliosis.

## 5. Differential Diagnosis

Many diseases can present with a sarcoidosis-like clinical phenotype (Table 8). Atypical manifestations that can make the diagnosis of sarcoidosis difficult are: hemoptysis, crackle rales, digital clubbing, unilateral hilar lymphadenopathy (3–5% of patients) or exclusive mediastinal lymphadenopathy without hilar lymph node enlargement, compressive lymphadenopathy, anterior mediastinal lymphadenopathy, miliary opacity, ground glass opacities, pleural involvement and bulky or cavitary mass (4%) on chest imaging and hypogammaglobulinemia [213,214]. After initial diagnosis, the involvement of a new organ is rare and even more in the absence of residual disease activity [215]. Such unusual circumstances should question the responsibility of sarcoidosis and force the clinician to repeat histological sampling and microbiological analyses, including standard stains and cultures. Accurate pathological examination is also crucial in making the right diagnosis, as neoplastic disorders and especially lymphoma which can mimic sarcoidosis [216].

### 5.1. Infectious Granulomatosis

Tuberculosis is one of the main differential diagnosis of sarcoidosis but fortunately, some biological and histological elements can help distinguishing these entities [217]. Interferon γ-release assay (IGRA) has the same sensitivity as in the general population since anergy does not affect its performances contrary to tuberculin skin test [218]. Of note, IGRA is useful for latent tuberculous infection but useless in active tuberculosis. Mycobacterial-specific polymerase chain reactions, despite their low sensitivity may be useful to differentiate mycobacterial infections from sarcoidosis [219,220]. Atypical mycobacteria and histoplasmosis are also classical infectious differential diagnoses as well as *Tropheryma whipplei* infection.

### 5.2. Other Inflammatory Diseases

Granulomatosis with polyangiitis (GPA) and granulomatosis with eosinophilia and polyangiitis (EGPA) are differential diagnosis to keep in mind when the lungs are affected. In inflammatory bowel diseases and more specifically Crohn’s disease, non-caseating granulomas are possible features all the more so as it could be associated with bronchiectasis [221].

### 5.3. Drug Induced Granulomatosis

Drug-induced sarcoid-like reactions (DISLR) are common during interferon therapy (especially with interferon alpha), which is known to induce granulomatous reactions with lung involvement in up to 76% of patients with hepatitis C and granulomatosis [222]. DISLR have also been described with TNF inhibitors and especially with soluble TNF-α receptor etanercept [223]. Immune checkpoint inhibitors, such as anti-cytotoxic T-lymphocyte-associated protein 4 (CTLA4) antibodies (ipilimumab) and anti-programmed cell death protein 1 (PD1) (nivolumab, pembrolizumab) or anti-PDL1 (ligand) antibodies (atezolizumab, durvalumab, avelumab) can provoke DISLR. BRAF and MEK inhibitors (dabrafenib, vemurafenib and trametinib, cobimetinib) are also associated with sarcoidosis [224]. SLR occurring during cancer immunotherapy are usually associated with paucisymptomatic lung, skin and mediastinohilar involvement.

### 5.4. Common Variable Immunodeficiency

Hypogammaglobulinemia is unusual in sarcoidosis and should alert the clinical for potential differential diagnosis with common variable immunodeficiency (CVID). Systemic granulomatosis can occur in CVID. Nevertheless, Bouvry et al. observed differences in clinical, biological and imaging features comparing CVID-related granulomatous disease (CVID-RGD) and sarcoidosis. Patients with CVID-RGD were more likely to present with hepato- and splenomegaly compared to sarcoidosis patients and had also more frequently history of recurrent infections and autoimmune diseases (especially cytopenia). CD4/CD8 ratio in BAL fluid was significantly higher in sarcoidosis than in CVID-RGD patients. Chest CT patterns are also different: nodules and peribronchovascular micronodules are more common in sarcoidosis whereas nodules with halo sign were most seen in CVID-RGD patients [225].

### 5.5. Neoplastic Disorders

One should not miss cancer diagnosis when confronted to granulomatosis, as a wrong diagnosis and the induced delay in properly treating the underlying malignancy could be deleterious [226]. Sarcoidosis may precede, follow, or occur concomitantly with many cancers, especially lymphomas, making the differential between them a diagnostic challenge. Both cancer and sarcoidosis are ^18^F-FDG avid. Therefore, the ^18^F-FDG PET CT may be useful in selecting possible biopsy sites (in priority, sites with the most important 18-FDG uptake), but not in distinguishing between both entities. 18F-3′-Fluoro-3′-deoxythymidine PET CT can help to distinguish malignant from non-malignant lesions but its role in cancer is to be determined [227]. If uncertainty remains, multiple biopsies and repeated tissue sampling may be necessary to evidence malignant cells.

### 5.6. Differential Diagnosis of Neurosarcoidosis

A special attention should be paid to the differential diagnosis of neurosarcoidosis (Table 9). MRI is the preferred imaging technique to differentiate sarcoidosis from its main mimickers and especially multiple sclerosis [161]. Three different initial presentation can be separated: suspected neurosarcoidosis in the course of histologically proven sarcoidosis, a sarcoidosis history without evidence of disease activity and the case of a patient without known sarcoidosis history [180]. In the latter situation, as well as in a context of inactive sarcoidosis, the evidence of active sarcoidosis should be assessed and histological examination has to be obtained as far as possible before initiating a systemic treatment. As seen before, ^18^F-FDG PET CT is of particular interest in those cases. If no evidence of active sarcoidosis can be found but the biopsy of any CNS structure reveals a sarcoid-like granulomatous reaction, one should be aware of the possible differential diagnosis between systemic neurosarcoidosis and a local sarcoid-like reaction, particularly related to an old neurological scar, neoplasia or an helminthic reaction [180].

## 6. Conclusions

Sarcoidosis is susceptible to encompassing numerous different clinical presentations. Whether it is symptomatic or not, or acute or not, sarcoidosis can involve variable organs with a diverse clinical impact from benign to very severe. The diagnosis of sarcoidosis is based on three main criteria: a compatible presentation, the evidence of non-caseating granulomas on histological examination and the exclusion of any alternative diagnosis. However, even when all these criteria are fulfilled, the probability of sarcoidosis diagnosis varies from “definite” to only “possible” depending upon the presence of more or less characteristic, depending on radio-clinical and histopathological findings and on the epidemiological context. The main differential diagnosis includes infections, especially tuberculosis, and malignancies, especially lymphoma. Several studies have recently highlighted the existence of distinct phenotypes of sarcoidosis, with a non-random distribution of organ involvement, which are associated with sex, geographical origin and socio-professional category. Currently, these phenotypes should be used to understand sarcoidosis not as a single disease but as various syndromes. Genetic and environmental determinants of such phenotypes have to be elucidated in future studies to better understand the pathophysiology of sarcoidosis and eventually to guide the treatment.

## Figures and Tables

**Figure 1 cells-10-00766-f001:**
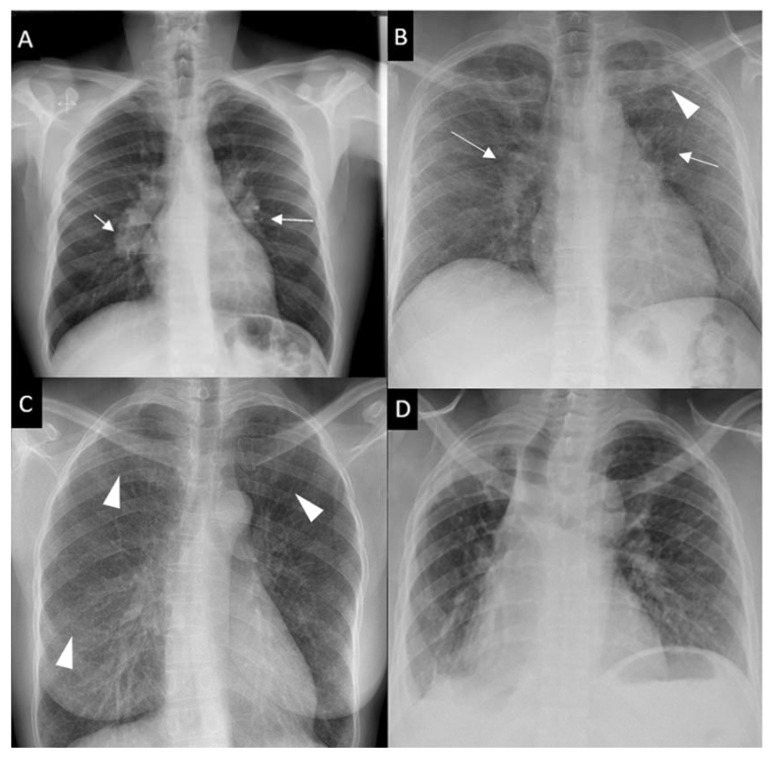
Chest x-ray (CXR) staging system. Stage 0-normal CXR (not shown); (**A**) Stage 1-bilateral hilar lymphadenopathy (white arrows); (**B**) Stage 2-bilateral hilar lymphadenopathy (white arrows) and pulmonary infiltrates in upper lobes (white arrowhead); (**C**) Stage 3-pulmonary infiltrates (white arrowhead) without bilateral hilar lymphadenopathy; (**D**) Stage 4-pulmonary fibrosis.

**Figure 2 cells-10-00766-f002:**
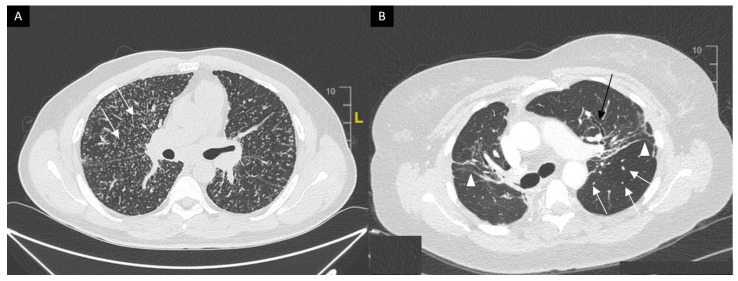
(**A**) Perilymphatic micronodules predominant in the right lung (white arrows). (**B**) Reticular opacities (white arrowheads), extensive traction bronchiectasis (black arrows) and perilymphatic nodules (white arrow) on lung windows. Findings consistent with sarcoidosis along with fibrosis.

**Figure 3 cells-10-00766-f003:**
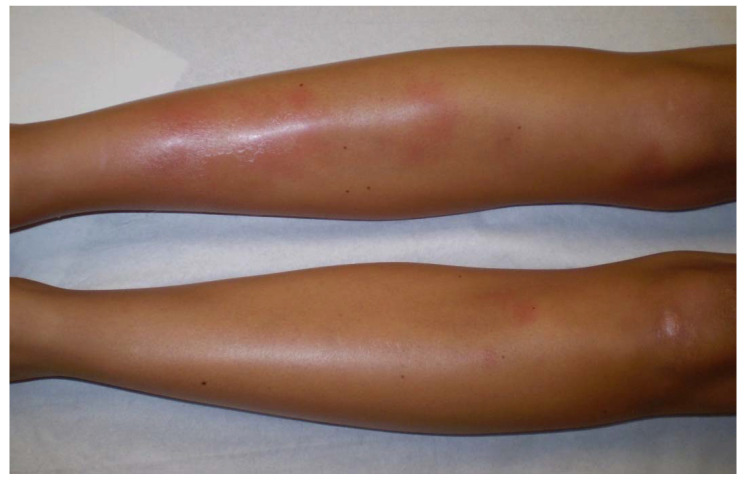
Erythema nodosum in a woman with Löfgren’s syndrome.

**Figure 4 cells-10-00766-f004:**
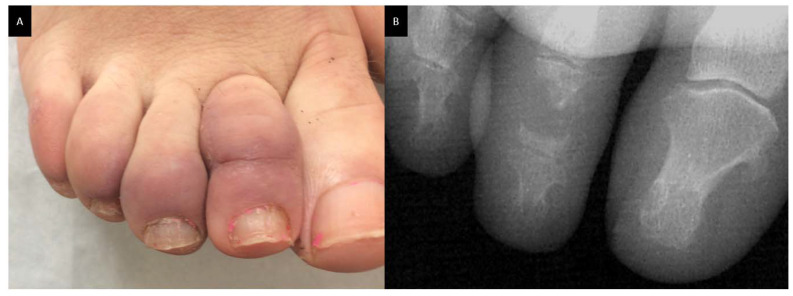
Dactylitis in a sarcoidosis patient; (**A**): skin on clinical examination; (**B**): X-ray on the same patient with punched out lesion of the second phalanx.

**Figure 5 cells-10-00766-f005:**
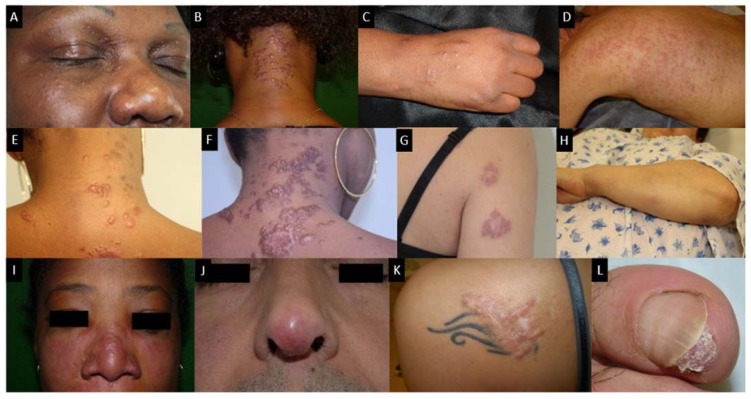
Skin sarcoidosis manifestations; (**A**–**C**): papular sarcoidosis, (**D**): diffuse maculopapular sarcoidosis; (**E**,**F**): Evolution of papular sarcoidosis into plaque sarcoidosis (same patient at a one-year interval); (**G**): annular plaque sarcoidosis; (**H**): subcutaneous sarcoidosis (Darier-Roussy type); (**I**): lupus pernio which has to be differentiated from (**J**): angiolupoid sarcoidosis (where telangiectasias are visible; (**K**): tatoo-sarcoidosis; (**L**): subungueal sarcoidosis.

**Figure 6 cells-10-00766-f006:**
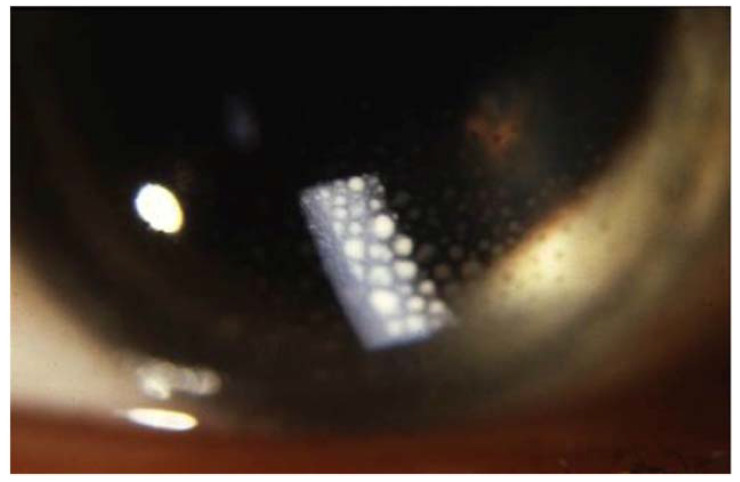
Large mutton-fat keratic precipitates (arrow) in sarcoidosis uveitis.

**Figure 7 cells-10-00766-f007:**
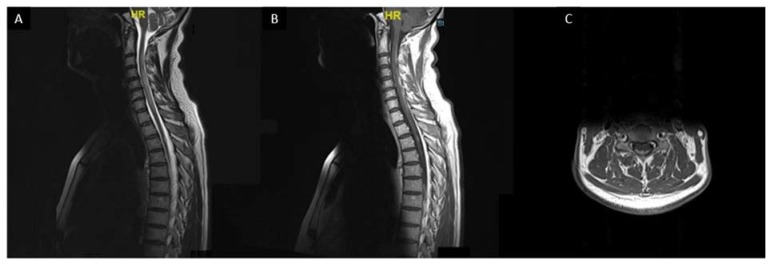
Sarcoidosis associated myelitis: (**A**): T2-weighted sagittal MRI slice showing longitudinally extensive T2 hyperintensity of the cervical and upper thoracic spinal cord associated with a focal T2 hyperintensity at T3 vertebra level; (**B**,**C**): T1-weighted post gadolinium sagittal (**B**) and axial (**C**) MRI images showing posterior subpial enhancement of the lesions.

**Figure 8 cells-10-00766-f008:**
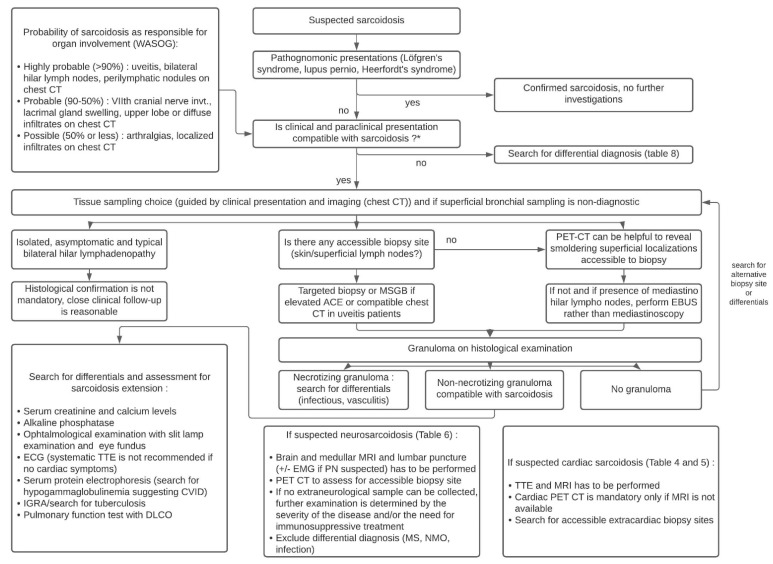
Diagnostic algorithm of sarcoidosis according to the ATS guidelines. Abbreviations: ACE: angiotensin converting enzyme; CT: computed tomography; CVID: common variable immunodeficiency; EBUS: endobronchial ultrasonography; EMG: electromyogram; KCO: organic carbon absorption coefficient; MRI: magnetic resonance imaging; MS: multiple sclerosis; MSGB: minor salivary glands biopsy; PET: positron emission tomography; PN: polyneuropathy; NMO: neuromyelitis optica; TTE: transthoracic echocardiography; WASOG: World Association of Sarcoidosis and other Granulomatous disorders.

**Figure 9 cells-10-00766-f009:**
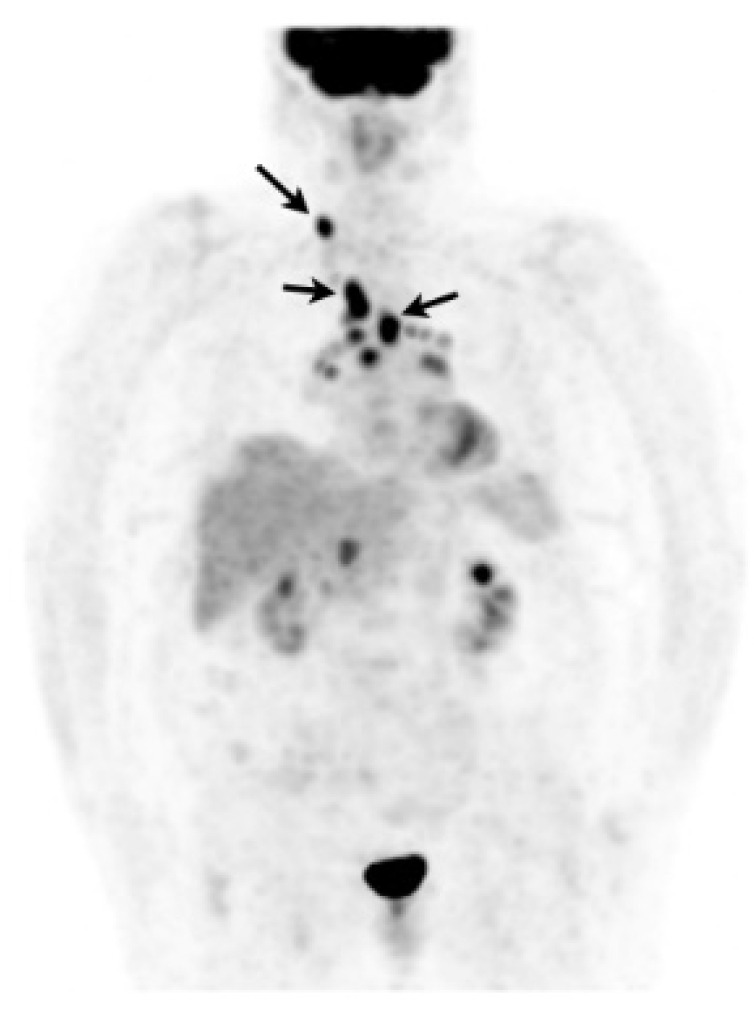
Caucasian woman (78 years old) suffering from bilateral unexplained panuveitis. Whole body ^18^F-FDG PET (maximum intensity projection anterior view) demonstrated significant bilateral hilar, mediastinal and right supraclavicular lymph nodes uptakes (black arrows) while there was no node enlargement on chest CT. A subsequent supraclavicular node dissection revealed noncaseating epithelioid granulomas consistent with sarcoidosis. Special staining and culture for mycobacteria were negative.

**Figure 10 cells-10-00766-f010:**
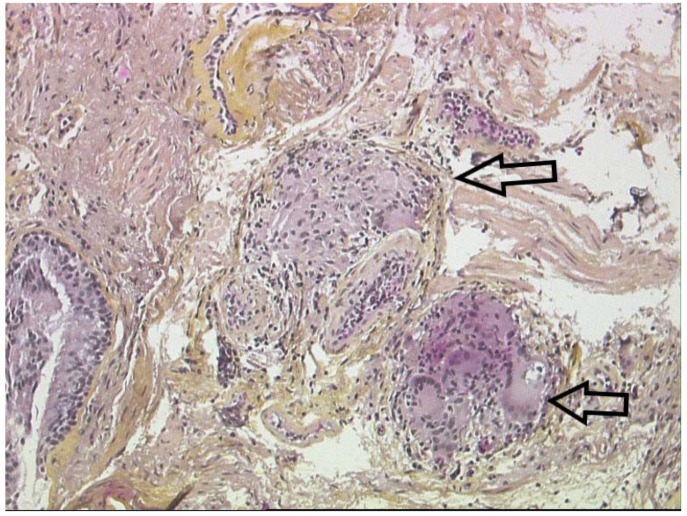
Lung biopsy: non-necrotizing epithelioid granulomas (arrows) with giant cells surrounding lymphocytes and fibrosis.

**Table 1 cells-10-00766-t001:** Common and uncommon specific cutaneous sarcoidosis findings (From [84,86]).

Common	Papules and papulonodules	Most common morphology of the specific cutaneous manifestations of sarcoidosis Description: numerous, firm, typically non-scaly, papular, usually smaller than 1 cm in size, Color: flesh-colored, yellow-brown, red-brown, purple-brown or hypopigmented Location typically on the face, especially on the eyelids and nasolabial folds, but also on the neck, trunk and extremities or within old scars Associated with a favorable prognosis
	Plaques	Description: oval or annular in shape, often well-demarcated, typically firm and scaly Color: red-brown to flesh-colored or purple-brown, sometimes yellow-brown Location: back, buttocks, face and extensor surfaces of the extremities; can arise within scars Associated with a chronic course
	Lupus pernio	Can be disfiguring; tends to affect African Americans and women disproportionately; associated with a chronic and refractory course, often requiring aggressive systemic treatment Description: smooth shiny plaques, which may become scaly Color: brown to violaceous or erythematous Location: centrofacial, especially on the nose, cheeks, lips, forehead, ears. Sarcoidosic involvement of the upper respiratory tract, bones (most commonly fingers and toes) and severe arthropathy are common.
	Subcutaneous Nodules	Description: firm, mobile, round to oval subcutaneous nodules or in the deep dermis, often with minimal surface changes Color: erythematous, flesh-colored, violaceous, or hyperpigmented Location: extremities, mainly upper extremities; trunk May be associated with benign systemic disease (debated)
Uncommon	Ichtyosiform	Description: fish scales, with adherent, polygonal, brown or white-gray scale Location: lower extremities
	Atrophic and ulcerative	Depressed plaques, easily ulcerated
	Mucosal	Buccal mucosa, gingiva, hard palate, tongue, posterior pharynx and salivary glands Papules, plaques, nodules and localized edema; papules or infiltrative thickening
	Erythroderma	Indurated, yellow-brown, red-brown, or purple-brown scaly plaques coalesce to involve large areas of skin, often with fine superficial scale or mild exfoliative dermatitis
	Alopecia	Scalp: scarring or nonscarring alopecia
	Nail sarcoidosis	Thinning, brittle and thickened nails, pitting, ridging, trachyonychia, hyperpigmentation, clubbing or pseudo-clubbing, destruction of the nail plate and scarring (e.g., pterygium nail)

**Table 2 cells-10-00766-t002:** Involvement of ocular structures and adnexa in sarcoidosis (except uveitis) (From [88,91,103,104,105,106,107]).

Location	Description
Lacrimal glands and Lacrimal drainage system (10–69%)	Often asymptomatic. Keratoconjunctivitis sicca (15–31%). Enlargement of the lacrimal glands is less frequent; the diagnosis can be made on Lacrimal gland biopsy.
Orbit	Women over 50. Diffuse orbital inflammation, usually unilateral, which can result in ptosis, limitations of ocular movements and diplopia.
	Ocular nerve palsy can occur from sarcoid involvement of the 3rd, 4th and 6th cranial nerves
Eyelid	Granuloma
Conjunctiva (6–40%)	Paucisymptomatic. Granuloma, conjunctivitis
Sclera (<3%)	Scleritis, episcleritis: diffuse inflammation, plaque or nodule; the diagnosis may be made with biopsy of a scleral nodule
Cornea	Interstitial keratitis (extremely rare).
Optic nerve (1–5%)	Optic neuropathy (++), granuloma, retrobulbar optic neuritis
	Predominantly Caucasian females. Frequently accompanied with uveitis and other findings of neurosarcoidosis. Prognosis is not favorable and permanent impaired visual acuity occurs in about one third of the patients. Sarcoidosis patients with opitc neuritis often experience a chronic course of the disease and steroid-sparing alternatives are commonly used.
Other neuro-ophthalmic manifestations	Rare: Horner’s syndrome, tonic pupil and optic-tract involvement

**Table 3 cells-10-00766-t003:** Common and uncommon neurological sarcoidosis findings and their evolution. From [156,158,161,164,169,173,178].

Common	Cranial nerves	VII (24%), II (21%) (see Table 2), V (12%), VIII (3-10%), oculomotor nerves, (2–6%), XI, XII (1%). Underlying mechanisms could be either epineural/perineural granulomatous inflammation of the nerve itself or granulomatous inflammation of the leptomeningeal compartment compressing the cranial nerves. Facial palsy is usually unilateral, but both sides can be affected at the same time in up to 30% [161]. Prognosis is usually good, with complete recovery in about 90% of patients under corticosteroids. Hearing loss is bilateral and asymmetrical in 75% of patients [179]. Half of the patients have abnormal vestibular testing and most of them have additional features of neurosarcoidosis (81%). Recovery occurs in most patients (70%).
	Meningeal	Clinical symptoms of meningeal irritation are seen in only 10% to 20% of patients with neurosarcoidosis. Many patterns of meningeal involvement have been described, but there is a tendency to basilar meninges involvement [164]. Leptomeningeal disease is a more severe disorder, with a risk of hydrocephalus and tissue destruction. Hydrocephalus is due to the obstruction of the ventricles by an inflammatory or granulomatous mass or infiltration of the meningeal spaces. CSF usually reveals mild monocyte pleocytosis (64%) and a protein elevation >1g/L (70%). Low glucose level (one-fifth of cases) and oligoclonal bands (30–42% of cases) are correlated with disability.
	Brain parenchyma	Intraparenchymal granulomatous lesions could be either a solitary mass or multiple nodules and may cause focal neurologic deficits, seizures, or increased intracranial pressure. Multiple non-enhancing white matter lesions are the most common imaging findings in sarcoidosis patients on MRI. These lesions are hyperintense on T2-weighted sequences and may be indistinguishable from those of multiple sclerosis.
Uncommon	Spinal cord	Most patients present with insidious, progressive, but non-specific sensory disorders, sphincter dysfunction and weakness over months before diagnosis [169]. Cervical (59%) > thoracic(29%) > conus (12%) involvement. On MRI, spinal cord sarcoidosis appears as heterogeneous patchy intramedullary lesions, a longitudinally extensive transverse myelitis with cord swelling and spinal cord atrophy in the late stages. 80% of patients will develop neurologic sequelae and 40% to 61% with a moderate to severe handicap.
	Pituitary	Hypogonadism is the most frequent endocrine disorder followed by TSH deficiency, diabetes insipidus and hyperprolactinemia [173]. MRI reveals infundibulum involvement (36.3%), pituitary stalk thickness (52%) and involvement of the pituitary gland (64%). MRI abnormalities can improve or disappear under corticosteroid treatment, but most endocrine defects are irreversible and require long-term substitutive opotherapy.
	Peripheral neuropathy	The most common form is chronic axonal sensory and/or motor neuropathy polyneuropathy [180]. Other forms include multiplex mononeuropathy, radiculopathy, brachial/lumbar plexitis, subacute demyelinating polyneuropathy mimicking Guillain-Barré syndrome, chronic demyelinating inflammatory neuropathy Epineural and perineural granulomas, as well as granulomatous vasculitis can cause ischemic axonal degeneration and demyelination resulting from local pressure. The prognosis is good for most patients after treatment, especially when there is a less severe presentation and a recent onset of symptoms [181] Small-fiber neuropathy: patients usually present with pain, burning sensation and paresthesia. These symptoms can be migratory and fluctuant. Dysautonomia causing orthostatic hypotension, palpitations, hyperhidrosis, gastrointestinal dysmotility, or bowel/bladder dysfunction is also observed in approximately half of patients. The diagnosis of small fiber neuropathy requires skin biopsy showing decreased intraepidermal nerve fiber density of lower than 5% of the population reference mean or quantitative sudomotor axonal reflex testing showing reduced sweat output [174].
	Stroke	Strokes which are thought to be related to sarcoidosis are ischemic (69%) or hemorrhagic (31%) [182] The main pathophysiological mechanism seems to be a granulomatous invasion of the vessels rather than cerebral vasculitis. Cardioembolic events (e.g., atrial fibrillation associated to cardiac sarcoidosis) and atherosclerotic lesions have to be excluded

Abbreviations: MRI: Magnetic Resonance Imaging.

**Table 4 cells-10-00766-t004:** Japanese Ministry of Health and Welfare criteria for diagnosing cardiac sarcoidosis [208].

Histological Diagnosis	Clinical Diagnosis
EMB revealed noncaseating granulomas and Histological or clinical diagnosis of extracardiac sarcoidosis	Presence of extracardiac sarcoidosis based on histological or clinical criteria plus either of the following: - ≥2 of the 4 major criteria - 1 major criteria and ≥2 minor criteria Major Criteria: - Advanced atrioventricular block - Decreased left ventricular ejection fraction, <50 (%) - Positive 67gallium uptake in the heart - Abnormal thinning of the basal interventricular septum Minor Criteria: - Abnormal ECG: ventricular arrhythmias, multifocal or frequent PVCs, complete RBBB, abnormal axis or Q waves - Abnormal echocardiogram: regional wall motion abnormality or morphological abnormality (aneurysm or wall thickening) - Nuclear imaging: perfusion defect on 201Thallium or 99mTechnetium single-photon emission computed tomography - Late gadolinium enhancement on cardiac magnetic resonance imaging - EMB: over moderate interstitial fibrosis and monocyte infiltration

Abbreviations: EMB: endomyocardial biopsy; ECG: electrocardiogram: PVC: premature ventricular complex; RBBB: right bundle branch block.

**Table 5 cells-10-00766-t005:** Heart Rhythm Society expert consensus statements on criteria for diagnosing cardiac sarcoidosis [138].

Histological Diagnosis	Clinical Diagnosis
EMB revealed noncaseating granulomas and Histological or clinical diagnosis of extracardiac sarcoidosis	It is probable cardiac sarcoidosis if: - There is presence of extracardiac sarcoidosis based on histological criteria And one or more of the following: - Treatment-responsive cardiomyopathy or heart block with corticosteroid +/-immunosuppressant drug - Unexplained decreased left ventricular ejection fraction <40 (%) - Unexplained sustained ventricular tachycardia; spontaneous or induced - Mobitz type II second-degree heart block or complete heart block - Patchy uptake of18F-flurodeoxyglycose on cardiac positron emission tomography, typical pattern consistent with CS - Late gadolinium enhancement on cardiac magnetic resonance imaging, typical pattern consistent with CS - Positive gallium uptake on scintigraphy, typical pattern consistent with CS AND: Other causes have been reasonably excluded

Abbreviations: CS: cardiac sarcoidosis; EMB: endomyocardial biopsy.

**Table 6 cells-10-00766-t006:** Consensus Diagnostic Criteria for Neurosarcoidosis From the Neurosarcoidosis Consortium Consensus Group [209].

Definite
The clinical presentation and diagnostic evaluation suggest neurosarcoidosis, as defined by the clinical manifestations and MRI, CSF and/or EMG/NCS findings typical of granulomatous inflammation of the nervous system after rigorous exclusion of other causes The nervous system pathology is consistent with neurosarcoidosis. Type a. Extraneural sarcoidosis is evident. Type b. No extraneural sarcoidosis is evident (isolated CNS sarcoidosis)
**Probable**
The clinical presentation and diagnostic evaluation suggest neurosarcoidosis, as defined by the clinical manifestations and MRI, CSF and/or EMG/NCS findings typical of granulomatous inflammation of the nervous system after rigorous exclusion of other causes There is pathologic confirmation of systemic granulomatous disease consistent with sarcoidosis
**Possible**
The clinical presentation and diagnostic evaluation suggest neurosarcoidosis, as defined by the clinical manifestations and MRI, CSF and/or EMG/NCS findings typical of granulomatous inflammation of the nervous system and after rigorous exclusion of other causes There is no pathologic confirmation of granulomatous disease.

Abbreviations: CNS: central nervous system; CSF: cerebrospinal fluid; EMG: electromyogram; MRI: magnetic resonance imaging; NCS: nerve conduction study.

**Table 7 cells-10-00766-t007:** Revised criteria of International Workshop on Ocular Sarcoidosis (IWOS) for the diagnosis of ocular sarcoidosis (from [211]).

**I. Other causes of granulomatous uveitis must be ruled out**
**II. Intraocular signs suggestive of ocular sarcoidosis**
1. Mutton-fat keratic precipitates (large or small) and/or iris nodules at pupillary margin (Koeppe) or in stroma (Busacca)
2. Trabecular meshwork nodules and/or tent-shaped peripheral anterior synechia
3. Snowballs/strings of pearls vitreous opacities
4. Multiple chorioretinal peripheral lesions (active and/or atrophic)
5. Nodular and/or segmental periphlebitis (+candle-wax drippings) and/or macroaneurysm in an inflamed eye
6. Optic-disc nodule(s)/granuloma(s) and/or solitary choroidal nodule
7. Bilaterality (assessed by ophthalmological examination including ocular imaging showing subclinical inflammation)
**III. Systemic investigations results in suspected ocular sarcoidosis**
1. Bilateral hilar lymphadenopathy by chest X-ray and/or chest computed CT scan
2. Negative tuberculin test in a BCG-vaccinated patient or interferon-gamma releasing assays
3. Elevated serum angiotensin converting-enzyme
4. Elevated serum lysozyme
5. Elevated CD4/CD8 ratio (>3.5) in bronchoalveolar lavage fluid
6. Abnormal accumulation of ^67^Ga scintigraphy or ^18^F-fluorodesoxyglucose positron emission tomography imaging
7. Lymphopenia
8. Parenchymal lung changes consistent with sarcoidosis, as determined by pneumologists or radiologists
**Diagnostic criteria of ocular sarcoidosis**
Diagnostic criteria of ocular sarcoidosis were worked out in 3 levels of certainty:
Definite ocular sarcoidosis: diagnosis supported by biopsy with compatible uveitis
Presumed ocular sarcoidosis: diagnosis not supported by biopsy, but bilateral hilar lymphadenopathy present with two intraocular signs
Probable ocular sarcoidosis: diagnosis not supported by biopsy and bilateral hilar lymphadenopathy absent, but three intraocular signs and two systemic investigations selected from two to eight are present

Abbreviations: CT: computed tomography.

**Table 8 cells-10-00766-t008:** Main differential diagnoses to exclude in sarcoidosis (from [216]).

Sarcoidosis Subtype	Possible Differential Diagnosis
Mediastino pulmonary sarcoidosis	Tuberculosis and mycobacterial infections
	Hodgkin’s disease and non Hodgkin’s lymphoma
	Histoplasmosis, coccidioidomycosis, aspergillosis
	Pneumoconiosis: chronic beryllium disease, titanium, aluminium, talc
	Hypersensitivity pneumonitis
	Drug reactions
	Granulomatosis with polyangiitis, granulomatosis with eosinophilia and polyangiitis
Extra thoracic sarcoidosis	Tuberculosis and mycobacterial infections
	Whipple disease, bartonellosis, Q fever, brucellosis, syphilis, toxoplasmosis, fungal infections
	Hodgkin disease and non Hodgkin’s lymphoma
	Tumor associated sarcoid reaction
	Crohn’s disease, primary biliary cirrhosis
	Drug induced sarcoidosis (interferon α and β, intravesical BCG therapy, TNFα inhibitors, immune check point inhibitors)
	Common variable immunodeficiency

**Table 9 cells-10-00766-t009:** Main differential diagnosis of neurosarcoidosis (from [180,228]).

**Infectious diseases** Tuberculosis Whipple’s disease Toxoplasmosis Histoplasmosis Toxocarosis Treponemal infections Lyme disease
**Granulomatous diseases (exclusion of infectious diseases)** Granulomatosis with polyangiitis Lymphomatoid granulomas
**Tumors** Neurolymphomas (e.g., Intravascular lymphoma) Histiocytic disorders (e.g., Erdheim–Chester disease) Gliomas Menineomas Leptmeningeal metastases
**Vasculitis** Behçet’s disease
**Systemic diseases** Lupus Sjôgren Amyloidosis Ig4-related disease
**Lymphocytic adenohypophysitis**
**Neurological diseases** Multiple sclerosis Neuromyelitis optica spectrum disorder spectrum and myelin oligodendrocyte antibody-induced demyelination Autoimmune/paraneoplastic encephalitis Susac syndrome Chronic lymphocytic inflammation with pontine perivascular enhancement responsive to steroids (CLIPPERS) Acute demyelinating encephalomyelitis Primary angiitis of the central nervous system

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
