# Peer review of "Sarcoidosis: A Clinical Overview from Symptoms to Diagnosis"

_cells, 2021, doi:10.3390/cells10040766_

Round 1

Reviewer 1 Report

The review article by Seve et al., summarizes broadly the problems connected with diagnosis of sarcoidosis. The manuscript is quite detailed with many examples and pictures illustrating the possible changes in different organs. There are no major drawbacks if this work. A few minor things are suggested to the authors:

A slight change in the title of section 4.1“Specific diagnosis criteria” could be made in order to specify the problematic diagnosis in cardiac, ocular and neurosarcoidosis. I’d also make more clear separation between these three parts in section 4.1. 

A few sentences in the introduction part need revision, like: “Sarcoidosis affects people of all ethnic backgrounds and occurs at any time in life but is slightly more frequent in African Americans and in Scandinavians more than in other Caucasian people”; “Patients with sarcoidosis have a lower survival rate compared with the general population [9], with a mortality rate of up to 7.6% [10].”

Author Response

Q: The review article by Seve et al., summarizes broadly the problems connected with diagnosis of sarcoidosis. The manuscript is quite detailed with many examples and pictures illustrating the possible changes in different organs. There are no major drawbacks if this work.

A: We thank the reviewer for his/her comments on our manuscript and his/her suggestions to improve its quality.

Q: A few minor things are suggested to the authors:

A slight change in the title of section 4.1“Specific diagnosis criteria” could be made in order to specify the problematic diagnosis in cardiac, ocular and neurosarcoidosis. I’d also make more clear separation between these three parts in section 4.1.

A: We changed the title of 4.1 section to “Specific diagnostic criteria in cardiac, neuro and ocular sarcoidosis”. We have also added three subsections entitled “4.1.1. Cardiac Sarcoidosis”, “4.1.2. Neurosarcoidosis”, “4.1.3. Ocular sarcoidosis”. Please find the changes respectively on lines 687, 688, 703 and 715.

Q: A few sentences in the introduction part need revision, like: “Sarcoidosis affects people of all ethnic backgrounds and occurs at any time in life but is slightly more frequent in African Americans and in Scandinavians more than in other Caucasian people”

A: This sentence has been modified to: “Sarcoidosis can occur regardless of ethnicity or age. However, African Americans and Scandinavians have a higher incidence of the disease than the rest of the Caucasian population. Sarcoidosis generally starts in adults under 50 years of age.”. Please find the modifications from lines 52 to 54.

Q: “Patients with sarcoidosis have a lower survival rate compared with the general population [9], with a mortality rate of up to 7.6% [10]”

A: This sentence has been modified to: “Patients with sarcoidosis have a shorter life expectancy than the general population [9]. The mortality ratio in the sarcoidosis patient population can, for example, exceed 25 per million in African American women [10]”. Please find the modifications from lines 75 to 77.

Reviewer 2 Report

I have read with great interest the manuscript entitled 'Sarcoidoisis: a clinical overview, from symptoms to the diagnosis.' By Seve et al.

This manuscript is well-written and very complete.

My comments the the authors are as follows:

MAJOR COMMENTS:

  • It might be useful to the reader to insert a section on general symptoms such as fatigue, concentration problems but also fever, weight los and night sweat. Especially fatigue is a very common (but not specific) symptom
  • Line 159-166: The section on BAL should also mention CD103/CD4 ratio with some references.  2008 Mar;126(3):338-44. doi: 10.1016/j.clim.2007.11.005. Epub 2008 Jan 8. Heron et al. Evaluation of CD103 as a cellular marker for the diagnosis of pulmonary sarcoidosis. Resp Med Diagnostic value of CD103 expression in bronchoalveolar lymphocytes in sarcoidosis. Mota PC et al

MINOR COMMENTS:

  • Figure 1: the infiltrates are not very clear in my print in B and C. Do you have a more illustrating example?
  • Line 146: The authors state that DLCO is a predictor of vascular envolvement (which is true), but it can also be a result of diffuse parenchymal envolvement or alveolitis. This should be stated.
  • Line 238: demaria et al does not have a ref number or reference in the list
  • Line 309: It might be helpfull to state that EN should preferably not be biopsied because it shows aspecific abnormalities and not granulomas
  • Line 487: SFN is very common and could possibly be elaborated a little bit further.
  • Figure 8: KCO is stated where in the paragraph lung involvement DLCO is stated

TYPO's

  • line 533: hypetrophic--> Hypertrophic
  • Table 8 HypersensitivitY pneumonitis
  • Line 723: GEPA: EGPA

Author Response

Q: I have read with great interest the manuscript entitled 'Sarcoidosis: a clinical overview, from symptoms to the diagnosis.' By Seve et al. This manuscript is well-written and very complete.

A: We thank the reviewer for his/her comments on our manuscript and his/her suggestions to improve its quality.

Q: My comments to the authors are as follows:

MAJOR COMMENTS:

It might be useful to the reader to insert a section on general symptoms such as fatigue, concentration problems but also fever, weight loss and night sweat. Especially fatigue is a very common (but not specific) symptom.

A: We agree that such a section could be helpful to the reader. A section called “2.1. General symptoms” has been added. Please find the modifications from line 96 to line 119:

“General symptoms are frequent in sarcoidosis [19]. For example, the prevalence of fatigue in sarcoidosis patients can reach 50 to 70% according to series [20]. Fatigue is not specific for sarcoidosis and it can be associated with different causes such as hypothyroidism, anxiety, depression, sleep apnea or active and severe inflammatory reaction [20]. A positive association was found between small fiber neuropathy and fatigue and also between dyspnea and fatigue [19]. The accurate detection of fatigue can be made with the Fatigue Assessment Scale [21]. The detection of fatigue is of utmost importance in sarcoidosis because on the one hand, fatigue is negatively related to quality of life in studies [22–25] and on the other hand the implementation of an adapted treatment can improve the patient's symptoms [20]. Concentration disturbances is also a frequently reported symptom in patients with sarcoidosis and may be due to various associated comorbidities or consequences of sarcoidosis (e.g.: sleep apnea and obstructive pulmonary disease). It is assumed that systemic treatment has a positive effect on sarcoidosis-associated cognitive complaint and cognitive disorders [19,26].

             Other non-specific constitutional symptoms in sarcoidosis can include fever and weight loss. In most cases, fever remains low grade but can sometimes reach 39 to 40°C [27]. In sarcoidosis, fever can be encountered during LS. Nevertheless, fever in a patient diagnosed with granulomatosis should rise the question of differential diagnoses and especially infectious differentials (e.g. tuberculosis). Sarcoidosis can also be a cause of fever of unknown origin and should be kept in mind when facing this clinical presentation [28]. Other general symptoms may be part of the initial clinical picture of sarcoidosis, such as weight loss and night sweats [27]. Of note, patients with hepatic manifestations of sarcoidosis can present with fever, night sweats and weight loss along with anorexia [29].”

Q: Line 159-166: The section on BAL should also mention CD103/CD4 ratio with some references.  2008 Mar;126(3):338-44. doi: 10.1016/j.clim.2007.11.005. Epub 2008 Jan 8. Heron et al. Evaluation of CD103 as a cellular marker for the diagnosis of pulmonary sarcoidosis. Resp Med Diagnostic value of CD103 expression in bronchoalveolar lymphocytes in sarcoidosis. Mota PC et al

 A: We do agree that this statement is relevant. We have added a mention to CD103 marker in BAL fluid in patients with sarcoidosis : “CD103+CD4+ T cells count as well as […] CD4 alveolitis is missing”. Please find this statement from line 195 to line 202.

MINOR COMMENTS:

Q: Figure 1: the infiltrates are not very clear in my print in B and C. Do you have a more illustrating example?

A: We do have a more illustrating example. We modified the figure accordingly to the reviewer’s advice. Please find the modification on page 4.

Q: Line 146: The authors state that DLCO is a predictor of vascular involvement (which is true), but it can also be a result of diffuse parenchymal involvement or alveolitis. This should be stated.

A: We agree that this statement was missing, and we have added it on lines 172 and 173: “Low DLCO value can also be the result of diffuse parenchymal involvement or alveolitis”.

Q: Line 238: Demaria et al does not have a ref number or reference in the list

A: We apologize for this missing information. Please find the correct reference on line 277 and in the References section : 18F-FDG PET/CT in bone sarcoidosis: an observational study, Clin Rheumatol, 2020, doi: 10.1007/s10067-020-05022-6.

Q: Line 309: It might be helpful to state that EN should preferably not be biopsied because it shows aspecific abnormalities and not granulomas

A: We agree that this should be added to the main text. Please find the added statement : “Histological examination is not warranted since histological structure of erythema nodosum is not specific to sarcoidosis and never granulomatous.” on line 337.

Q: Line 487: SFN is very common and could possibly be elaborated a little bit further.

A: We do agree that sarcoidosis associated SFN should be detailed a little bit further. We added some statements relative to SFN diagnostic tools and clinical presentation. Please find the modifications “SFN symptoms […] reproductible questionnaire” from line 529 to line 537. 

Q: Figure 8: KCO is stated where in the paragraph lung involvement DLCO is stated

A: This correction has been added on figure 8.

Q: TYPO's

line 533: hypetrophic--> Hypertrophic

Table 8 HypersensitivitY pneumonitis

Line 723: GEPA: EGPA

A: These typos have been corrected at their respective locations in the main text and in Table 8.

We would also like to alert the reviewer that we have added the term “positive” on line 148 which was missing.

Reviewer 3 Report

This is a very comprehensive and well written manuscript about sarcoidosis

I have some minor questions and remarks about the manusript

line 103: I would also mention the term Scadding stages because this is also widely used so the reader knows that both Siltzbach classification and Scadding stages are the same.

line 111:  it would not use the term "high" with regard to its prognostic value.  This is not very high in my opnion

line 149: please clarify CPI and why this is more predictive than FVC (because implementation of diffusion capacity in the CPI)

line 405. In light of the rest of the paragraph, I would prefer to use Cardiac Sarcoidosis instead of Heart Sarcoidosis

line 411: I would recommend to use the abbreviation CS instead of CaS. In other papers the abbreviation CS is most commonly used.

line 427. Please clarify that a normal TTE does not exclude CS

line 706-710: the authors state to repeat histological and microbiological sampling. And what about annual ECG to screen for Cardiac Sarcoidosis. It think this should be mentioned as well.

line 714: the authors should state clearly that IGRA detects LTBI and not clinical active tuberculosis infection. 

Author Response

This is a very comprehensive and well written manuscript about sarcoidosis

A: We thank the reviewer for his/her comments on our manuscript and his/her suggestions to improve its quality.

Q: I have some minor questions and remarks about the manuscript

line 103: I would also mention the term Scadding stages because this is also widely used so the reader knows that both Siltzbach classification and Scadding stages are the same.

A: We have modified the classification name from Siltzbach to Scadding. Please find the modification on line 127 : “Scadding’s classification defines five stages of sarcoidosis on a CXR (Figure 1).”

Q: Line 111:  it would not use the term “high” with regard to its prognostic value.  This is not very high in my opinion

A: We have suppressed the term “high” in this sentence. Please find the modification on line 135.

Q: line 149: please clarify CPI and why this is more predictive than FVC (because implementation of diffusion capacity in the CPI)

A: This is a very interesting statement to add to the main text. We have modified the manuscript according to the reviewer’s recommendations. Please find the modifications “The Composite Physiologic Index (CPI) is a composite score calculated with disease extent on CT, DLCO, FVC and FEV1 values and is more closely correlated to disease extent in idiopathic pulmonary fibrosis than isolated parameters from PFTs such as FVC and DLCO [33]. CPI is also more correlated to disease extent than FVC because of DLCO implementation in the calculated index. The CPI when >40 is predictive of mortality in sarcoidosis patients [32].” from line 173 to 178.

Q: line 405. In light of the rest of the paragraph, I would prefer to use Cardiac Sarcoidosis instead of Heart Sarcoidosis

A: We have modified the title of this section to : “Cardiac Sarcoidosis”. Please find the modification on line 443.

Q: Line 411: I would recommend using the abbreviation CS instead of CaS. In other papers the abbreviation CS is most commonly used.

A: We do agree that this abbreviation is more suitable. We have modified all the occurrences of CaS and replaced them with CS.

Q: line 427. Please clarify that a normal TTE does not exclude CS

A: We do agree that this statement is fundamental in cardiac sarcoidosis management. Please find the modification on line 468-9 : “Of note, a normal TTE does not exclude CS.”

Q: line 706-710: the authors state to repeat histological and microbiological sampling. And what about annual ECG to screen for Cardiac Sarcoidosis. It think this should be mentioned as well.

A: This statement was added in the “Cardiac Sarcoidosis” section. Please find the modification on line 455-7 : “Current ATS guidelines recommend that electrocardiogram should not be performed every year in any patient with known sarcoidosis since clinically silent CS usually follows a benign course”.

Q: line 714: the authors should state clearly that IGRA detects LTBI and not clinical active tuberculosis infection.

A: We have clarified this statement. Please find the modification on line 769-70 : “Of note, IGRA is useful for latent tuberculous infection but useless in active tuberculosis”.